# Prognostic Value of High-Sensitivity Cardiac Troponin in Women

**DOI:** 10.3390/biom12101496

**Published:** 2022-10-17

**Authors:** Giandomenico Bisaccia, Fabrizio Ricci, Mohammed Y. Khanji, Giulia Gaggi, Andrea Di Credico, Sabina Gallina, Angela Di Baldassarre, Barbara Ghinassi

**Affiliations:** 1MIUR Department of Excellence, Department of Neuroscience, Imaging and Clinical Sciences, University “G. d’Annunzio” of Chieti-Pescara, Via Luigi Polacchi, 66100 Chieti, Italy; 2Department of Clinical Sciences, Lund University, E-205 02 Malmö, Sweden; 3Newham University Hospital, Barts Health NHS Trust, Glen Road, Plaistow, London E13 8SL, UK; 4Barts Heart Centre, Barts Health NHS Trust, West Smithfield, London EC1A 7BE, UK; 5NIHR Barts Biomedical Research Centre, William Harvey Research Institute, Queen Mary University, London EC1A 7BE, UK; 6Reprogramming and Cell Differentiation Lab, Center for Advanced Studies and Technology (CAST), 66100 Chieti, Italy; 7Department of Medicine and Aging Sciences, University “G. d’Annunzio” of Chieti-Pescara, 66100 Chieti, Italy

**Keywords:** cardiac troponins, prognosis, cardiovascular disease, cardiovascular prevention, sex differences, women’s health

## Abstract

High-sensitivity cardiac troponin assays have become the gold standard for diagnosing acute and chronic myocardial injury. The detection of troponin levels beyond the 99th percentile is included in the fourth universal definition of myocardial infarction, specifically recommending the use of sex-specific thresholds. Measurable concentrations below the proposed diagnostic thresholds have been shown to inform prognosis in different categories of inpatients and outpatients. However, clinical investigations from the last twenty years have yielded conflicting results regarding the incremental value of using different cut-offs for men and women. While advocates of a sex-specific approach claim it may help reduce gender bias in cardiovascular medicine, particularly in acute coronary syndromes, other groups question the alleged incremental diagnostic and prognostic value of sex-specific thresholds, ultimately asserting that less is more. In the present review, we aimed to synthesize our current understanding of sex-based differences in cardiac troponin levels and to reappraise the available evidence with regard to (i) the prognostic significance of sex-specific diagnostic thresholds of high-sensitivity cardiac troponin assays compared to common cut-offs in both men and women undergoing cardiovascular disease risk assessment, and (ii) the clinical utility of high-sensitivity cardiac troponin assays for cardiovascular disease prevention in women.

## 1. Introduction

Cardiovascular disease (CVD) is a major cause of morbidity and mortality worldwide, accounting for an estimated 18.6 million deaths in 2019 [1]. Preventative strategies are recommended by international guidelines to reduce the burden of CVD in the general population, along with its social, clinical, and financial costs [2]. So far, most of the scientific evidence in the cardiovascular literature has been developed with an underrepresentation of women, in both experimental and clinical contexts, despite growing knowledge on significant sex-based biological differences in the cardiovascular system [3,4,5,6]. The use of clinically meaningful biomarkers may help us to reduce the gap between men and women in the field of cardiovascular medicine and research [7]. Cardiac biomarkers are objectively measured features of physiological and disease-related processes with a plethora of possible applications, ranging from disease diagnosis to staging, prognosis, and the prediction of response to interventions [8,9]. Among all proposed cardiac biomarkers, cardiac troponins and natriuretic peptides have emerged as some of the most powerful and cost-effective clinical tools to diagnose and risk-stratify patients with suspected CVD [10,11,12]. In 2018, the European Society of Cardiology, American College of Cardiology, American Heart Association, and World Heart Foundation issued the fourth universal definition of myocardial infarction (MI), defining MI as a rise and/or fall in cardiac troponin with at least one value above the 99th percentile upper reference limit (URL) [13]. Although its main clinical use lies in the detection of patients with acute coronary syndromes, in recent years, the widespread clinical availability of high-sensitivity cardiac troponin testing and the subsequent detection of low-grade troponin elevation in patients without acute coronary syndromes [14] have determined an increasing interest in deploying cardiac troponins as potential prognostic markers in the general population. The more widespread use of high-sensitivity assays in the general population has opened the door to the detection of measurable troponin concentrations below the 99th percentile, which have been shown to predict all-cause mortality [15] and major cardiovascular events [14,16]. The use of high-sensitivity cardiac troponin assays was recently recommended for CV risk prognostication in the general population [17,18]. The possible mechanisms underlying the observed associations have been previously investigated [19,20,21]. The presence of detectable troponin levels has been linked to subclinical myocardial injury, and higher troponin concentrations have been associated with a higher prevalence of underlying coronary artery disease and left ventricular hypertrophy. However, the potential role of cardiac troponin assays for cardiovascular disease prevention in the general population is not fully understood. Further knowledge in this field is needed, particularly for CVD prevention in women, who are likely to face significant gender bias in cardiac testing [22,23].

## 2. Cardiac Troponins: A Translational Overview

The cardiac troponin complex is part of the cardiomyocyte contractile apparatus. It comprises three proteins, cTnI, cTnT, and cTnC, enabling the regulation of cardiac excitation–contraction coupling [24]. The release of cardiac troponins from the cardiomyocyte may occur due to different pathophysiological processes [25]. Established causes of troponin secretion include inflammation, necrosis, apoptosis, exocytosis, and physiological myocyte turnover. Since troponin C in the cardiomyocyte has an identical structure to its skeletal muscle counterpart, the elevation of the serum levels of this protein is not specific to cardiac disease. As such, myocardial injury is ascertained if detectable cardiac troponin concentrations are found above the 99th percentile of the upper reference limit (URL) [13]. An acute cardiac event is to be suspected when cardiac troponin levels exhibit a rise-or-fall pattern. In a clinical scenario suggestive of overt ischemia (e.g., typical ischemic chest pain and ischemic electrocardiographic changes), myocardial injury may be adjudicated as acute myocardial infarction; instead, in the absence of clinical signs and symptoms of ischemia, troponin elevation could be due to either chronic release or acute, nonischemic myocardial injury [26]. Noncommunicable diseases associated with continuous troponin release include chronic heart failure, cardiomyopathies, hypertensive heart disease, and chronic kidney disease. Acute nonischemic injury is instead found in patients with myocarditis, acute heart failure, and systemic diseases, including sepsis and pulmonary embolism. 

Although the diagnostic threshold for myocardial injury lies at the 99th percentile URL, detectable concentrations below the proposed cut-off have been shown to confer prognostic information in the general population [15]. A meta-analysis of 28 studies involving 154,052 participants without baseline cardiovascular disease (CVD) found that individuals in the top tercile of the normal cardiac troponin distribution have a significantly increased risk of future CVD events [14]; more recently, a meta-analysis of 24 studies comprising 203,202 community-dwelling individuals demonstrated a significant association between elevated cardiac troponin concentrations and cardiovascular events, heart failure hospitalization, and cardiovascular as well as all-cause death [27]. It has been further suggested that the individual’s temporal trajectory of cardiac troponin levels may be additionally informative of future patient outcomes [28]. Cardiac troponin levels were shown to reflect the left ventricular mass index [19,20], as well as the presence of significant coronary artery disease [20]. Significantly, a cohort study enrolling 880 women tested with a high-sensitivity assay found cardiac troponin I to be higher in patients with pregnancy-induced hypertension or preeclampsia [29]. There is evidence to suggest that such increased levels may normalize over time in women with controlled hypertension [30]. Moreover, cardiac troponin levels were shown to provide accurate prognostic information in community-dwelling individuals with diabetes [31] and chronic heart failure [32,33], as well as in patients with left ventricular hypertrophy [34] and valvular heart disease [35]. Interestingly, a genome-wide association study on Scottish outpatients found that genetic causes of increased serum levels differ between cardiac troponin I and T, with cTn-I being a more specific marker of cardiovascular disease [36]. 

## 3. Why Is Troponin an (Almost) Ideal Biomarker?

The measurement of cardiac troponin concentrations offers several unique advantages over other traditional cardiac biomarkers. Cardiac troponins are tissue-specific, are highly stable in vitro, and can be obtained from both plasma and serum; moreover, cardiac troponin assays are inexpensive, fast, and highly reliable and have been shown to outperform historical gold-standard tests in the diagnosis of acute myocardial infarction [37,38,39]. From a technical standpoint, high-sensitivity cardiac troponins have a very low limit of detection and offer the benefit of very little biological variability [18]. Moreover, these assays have a high degree of individuality [40], allowing for serial results to correctly reflect subtle changes in troponin secretion.

## 4. The Case for Sex-Specific Troponin Cut-Offs

Sex-based differences in the cardiovascular system have long been established [41,42]. Normal reference values of cardiac dimensions are greater in men than in women across all age groups, both at echocardiography and on cardiovascular magnetic resonance imaging and despite adjustment for body surface area [43,44]. Epidemiological studies also show sex-related differences in CVD prevalence and features [4]. Women with acute coronary syndromes still represent a clinical challenge: they are less frequently diagnosed, often present with atypical symptoms, have a greater burden of non-obstructive disease, and suffer higher morbidity and mortality compared with men. Structural flaws in the management of women with ischemic heart disease have been highlighted [4,23]. In this context, the use of sex-specific 99th percentile URLs for high-sensitivity troponin assays has been proposed to reduce treatment disparities [45,46] and was recommended in the fourth universal definition of MI [13], although uncertainty persists regarding the additional value of a sex-specific approach [47].

The notion that sex-based differences may exist in the mechanisms of release, diagnostic yield, and prognostic significance of cardiac troponins is not novel, and there has been an ongoing debate in the medical community regarding the most appropriate approach to the clinical use of these biomarkers. Different cohorts of apparently healthy individuals have demonstrated that men have higher levels than women [7,48,49,50]. This finding has been related to genotypic and phenotypic differences between sexes [51]. Men have higher indexed ventricular mass, cardiomyocyte volume, and protein content and may be exposed to higher shear stress due to significantly higher blood pressure levels. Cardiac inflammatory, pro-fibrotic, and apoptotic pathways show sexual dimorphisms as well, partly explained by hormonal modulation [52,53]. However, initial efforts at characterizing sex-specific cardiac troponin levels in unstable coronary patients did not show significant differences between sexes. A Swedish study published in the year 2000 consisting of 1455 such patients (36% females) demonstrated the equivalent prognostic performance of cardiac troponin concentrations in men and women with the use of contemporary standard-sensitivity assays [54]. However, sexual dimorphism in cardiac troponin levels was subsequently reported, with higher cardiac troponin I in males demonstrated in patients with acute myocardial injury [55]. Higher 99th percentile values for cardiac troponin I were also demonstrated in men in a cohort of apparently healthy adults [56], a finding later confirmed [16,57]. A higher troponin T concentration was also observed in community-dwelling adult males [19], as well as in the elderly [58]. With regard to prognostic significance, a cardiac troponin I cut-off of >7.6 ng/mL was specifically identified for the prediction of major adverse cardiovascular events in postmenopausal women undergoing cardiac surgery [59], and the leveraging of sex-specific cut-offs was advocated [60]. At the same time, the diagnostic accuracy of baseline troponin levels to detect myocardial infarction was found to be independent of sex in patients from the CHECKMATE cohort [61], again calling into question the real-world necessity for the use of a sex-based approach [62]. However, the studies outlined so far have based their investigations on standard sensitivity cardiac troponin assays; the advent of high-sensitivity assays eventually shuffled the cards [7,15,48,63,64]. With the new assays, the prevalence of detectable cardiac troponin levels increased from <5% (standard assays) to >80% (high-sensitivity assays) in healthy individuals. 

Male sex was confirmed as a predictor of high-sensitivity cardiac troponin I and T concentrations in the elderly [65,66], and further evidence was provided that the use of sex-specific cut-offs was superior to a sex-neutral cut-off in patients suspected of having acute coronary syndromes [67]. Moreover, high-sensitivity cardiac troponin I levels were reported to better predict all-cause mortality among females presenting to the emergency department [68] and to better prognosticate in women with non-ST-elevation acute coronary syndromes [48], contradicting previous evidence obtained with use of standard-sensitivity assays [54] and with fourth-generation assays [69]. The results of the UTROPIA study confirmed the higher prognostic yield of sex-specific cut-offs to predict adverse cardiovascular events in emergency department patients [70]. Further, a study powered to identify sex- and age-specific cut-offs for cardiac troponin T with the use of a high-sensitivity assay demonstrated the poor diagnostic yield of a uniform 14 ng/mL cut-off in both sexes and across multiple age groups [71]. Despite growing evidence that sex-specific cut-offs add to the prognostication of women in the general population, the incremental benefit of a sex-specific approach, with the use of personalized thresholds for men and women, over the use of a single multivariable model adjusted for sex is uncertain, since no head-to-head comparison studies for these two approaches are available to date.

## 5. Cardiac Troponins and Prognosis of Community-Dwelling Women

To date, the prognostic yield of high-sensitivity cardiac troponins for the cardiovascular risk stratification of women in the general population has been investigated in several studies (Table 1). 

In the Women’s Health Study, serum levels of cardiac troponin T were measured with the use of a high-sensitivity assay in two separate cohorts, diabetic and non-diabetic female health professionals, with a total sample size of 1064 females who were followed up for 12 years [72]. The authors found that the prevalence of detectable troponin T was significantly higher in women with diabetes (45% versus 30%), and detectable levels were associated with cardiovascular outcomes in diabetic women only, suggesting subclinical myocardial injury in these patients. In 2014, the results of the FINRISK study on 7899 participants without baseline CVD were published [64]. In this cohort, the use of high- and super-sensitivity cardiac troponin I assays was associated with significantly increased predictive ability for cardiovascular events, including incident heart failure and myocardial infarction, even after adjustment for traditional risk factors, C-reactive protein, and N-terminal pro-brain natriuretic peptide. However, no sex differences were demonstrated. 

A large study of 15,340 Europeans (51% females) tested with a high-sensitivity cardiac troponin I assay for the prediction of future cardiovascular events found a lower threshold for high risk in women, namely, 4.7 pg/mL, as compared with 7.0 pg/mL for men [16]. In this study, with a very long follow-up period of 20 years, the authors found that the application of a sex-specific approach resulted in substantial risk reclassification, with the additional identification of individuals at high risk of future events. More recently, a Scottish study reported on the prognostic significance of the serum levels of cardiac troponins I and T as assessed by high-sensitivity troponin assays in 19,501 adult individuals (58% women) followed up for a median of 7.8 years. The cohort had a low prevalence of CVD at baseline (3.2% in women and 6.3% in men). They found that cardiac troponin levels were stronger predictors of future cardiovascular events in women [73]. 

A stronger association between cardiac troponin T levels and the risk of incident heart failure was reported in women from the Atherosclerosis Risk in Communities (ARIC) study [32]. Subsequently, it was independently demonstrated, in patients from the same cohort, that the addition of time trajectories of cardiac troponin T to an adjusted model comprising traditional risk factors, N-terminal pro–brain natriuretic peptide, and baseline cardiac troponin T provided additional information on the future risk of incident heart failure, as well as on incident coronary disease and death [28]. In this study, the use of sex-specific cut-offs did not further improve the model performance. In 2019, more results from the ARIC were published for individuals without baseline CVD who had high-sensitivity cardiac troponin I measured [74]. This study confirmed the stronger association of cardiac troponin levels with cardiovascular outcomes in women, including for cardiac troponin I, and showed that cardiac troponins were complementary to one another in the prediction of future events.

**Table 1 biomolecules-12-01496-t001:** Main studies on the sex-specific prognostic significance of high-sensitivity cardiac troponin assays in outpatient populations.

Year	Study	Target Population	Country	Sample Size (n)	Female Sex	Cardiac Troponin Assay (s)	Main Outcome	Follow-Up	Main Findings
2021	Generation Scotland [73]	General population	Scotland	19,501	58%	Roche Elecsys hs-TnT Abbott Architect STAT hs-TnI	CV events	7.8 years	HR of 9.7 in women (compared to 5.6 in men) when a threshold of 10 ng/L was applied for hs-TnI
2019	ARIC [74]	Participants free of CVD at baseline	United States	8121	58%	Abbott Architect STAT hs-TnI	CV events	15 years	Stronger association of TnI and CHD in women (HR of 1.54 in women versus 1.29 in men)
2017	Busselton Health Study [75]	General population	Australia	3939	57%	Abbott Architect STAT hs-TnI	AF hospitalization	20 years	Similar prognostic value between sexes (aHR of 1.22 in women and 1.17 in men)
2016	ARIC [28]	General population	United States	8838	59%	Roche Elecsys hs-TnT	CHD, HF, all-cause mortality	14 years	Clinical equipoise between common and sex-specific approaches
2016	AGES-Reykjavik [76]	Community-dwelling elderly	Iceland	5764	58%	Abbott Architect STAT hs-TnI	CV events, all-cause mortality	8.9 years	Similar prognostic value between sexes (aHR of 1.27 in women and 1.23 in men for all-cause death)
2015	ActiFE Ulm [77]	Community-dwelling elderly	Germany	1422	43%	Roche Elecsys hs-TnT Abbott Architect STAT hs-TnI	All-cause mortality	4 years	Higher prognostic value in women (aHR of 3.33 in women and 1.92 in men for hs-TnI)
2015	HUNT [78]	General population	Finland	9710	54%	Abbott Architect STAT hs-TnI	CV mortality	6.4 years	Higher prognostic value in women (aHR of 1.44 in women versus 1.10 in men)
2015	HUNT [79]	General population	Finland	9114	55%	Abbott Architect STAT hs-TnI	Incident HF, incident MI (composite)	13.9 years	Higher prognostic value in women (aHR of 1.50 in women versus 1.27 in men)
2014	MORGAM [16]	General population	Scotland	15,340	51%	Abbott Architect STAT hs-TnI	CV events	20 years	Lower threshold for high risk in women (aHR of 1.35 in women versus 1.26 in men; threshold of 4.7 ng/mL and 7.0 ng/mL in women and men, respectively)
2014	FINRISK [64]	Participants free of CVD at baseline	Finland	7899	50%	Abbott Architect STAT hs-TnI Abbott Architect STAT cs-TnI Singulex Erenna Cardiac ss-TnI	CV events, incident HF, incident MI, all-cause death	14 years	Similar prognostic value between sexes, except for MI (aHR of 1.19 in women versus 1.17 in men), HF (aHR of 1.12 in women as compared with 1.24 in men), and all-cause death (aHR of 1.11 in women versus 1.03 in men)
2013	ARIC [32]	General population	United States	9868	66%	Roche Elecsys hs-TnT	Incident HF	10.4 years	Higher prognostic value in women (HR of 5.3 in women and 4.3 in men with hs-TnT values > 25 ng/L)
2011	Women’s Health Study [72]	Women free of CVD at baseline	United States	1076	100%	Roche Elecsys hs-TnT	CV events, CV mortality	12.3 years	Prognostic significance in diabetic, but not non-diabetic, women

AF, atrial fibrillation; aHR, adjusted hazard ratio; CHD, coronary heart disease; CV, cardiovascular; CVD, cardiovascular disease; HF, heart failure; HR, hazard ratio; hs-TnI, high-sensitivity cardiac troponin I; hs-TnT, high-sensitivity cardiac troponin T.

In the HUNT study, a cohort of 9712 Norwegian outpatients was followed up for a median of 6 years, and their high-sensitivity cardiac troponin I levels were obtained at baseline [78]. In the cohort, men had a higher crude rate of cardiovascular death, but this effect was no longer present when adjusting for cardiac troponin I concentrations. An overall stronger association between cardiac troponin I levels and cardiovascular death was reported for women. A subsequent analysis targeted at the prediction of myocardial infarction and incident heart failure was published for this cohort study, where the authors showed that cardiac troponin I was more strongly associated with these outcomes in women [79]. Cardiac troponin levels were also found to better predict all-cause mortality in asymptomatic elderly women as compared with age-matched men [77], although this finding was not confirmed in the AGES-Reykjavik cohort [76]. Finally, cardiac troponin I was also found to predict atrial-fibrillation-related hospitalizations, but no effect of sex was found in a cohort of Australian community-dwellers [75].

## 6. Future Perspectives

Several studies published in the last decade have shown significant sex differences in serum levels of cardiac troponins as assessed with high-sensitivity assays, and the detection of serum levels well below the 99th percentile URL has demonstrated excellent prognostic yield in the general population. However, most cardiovascular risk prediction models specifically developed for women do not include cardiac troponins or other surrogates of myocardial damage [80], and cardiac troponin testing is not implemented in daily practice. Another fascinating aspect is the use of cardiac troponins as therapeutic targets. Despite evidence that troponin level lowering is associated with an overall reduction in CV risk in men [81], and that it accurately predicts the therapeutic response in high-risk populations [82], there is currently a lack of such evidence in women. Future research should fill the current knowledge gaps by investigating the molecular pathways involved in the release of cardiac troponins in healthy and diseased cardiomyocytes in basic and translational studies and fully incorporate biological sex fingerprints in clinical studies.

## 7. Conclusions

In this review, we have synthesized data on the role of cardiac troponins for cardiovascular risk stratification in women, with a focus on community-dwelling populations. Women are still underrepresented in cardiovascular research and underdiagnosed in clinical practice, although progress has been made in the last decade to reduce implicit bias in cardiovascular testing. Evidence is accumulating that high-sensitivity cardiac troponin assays can help leverage gender differences in cardiovascular disease due to the high prognostic yield in ambulatory populations, particularly in women. Cardiovascular practitioners should be encouraged to leverage a sex-based approach to the diagnosis, prognostication, and treatment of cardiovascular disease. Whether this approach should encompass the use of cardiac troponins in the general population is still debated.

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
