# Peer review of "Prognostic Value of High-Sensitivity Cardiac Troponin in Women"

_biomolecules, 2022, doi:10.3390/biom12101496_

Round 1

Reviewer 1 Report

The study is well conducted and performed.

Please, include authors for reference 7.

Please, include a section of methods with the description of the bibliographic search and inclusion and exclusion criteria.

Please, discuss the relevance of including specific sex-cutoff points instead of multivariate models adjusted by sex.

Author Response

GENERAL COMMENT

The study is well conducted and performed.

Author’s Reply: Thank you for taking the time to review our manuscript.

SPECIFIC COMMENT

  1. Please, include authors for reference 7.

Author’s reply: Item verified and modified accordingly.

  1. Please, include a section of methods with the description of the bibliographic search and inclusion and exclusion criteria.

Author’s reply: Item verified. Manuscript now includes a section 8. Materials and Methods detailing the literature search methodology.

  1. Please, discuss the relevance of including specific sex-cutoff points instead of multivariate models adjusted by sex.

Author’s reply: Item verified and manuscript modified accordingly. See lines 185-190.

Reviewer 2 Report

To the Authors

General Considerations

The aim of this review is to summarize our current understanding of sex-based differences in cardiac troponin levels.  The two most important issues discussed by Authors are: 1- the prognostic significance of sex-specific diagnostic thresholds of high-sensitivity cardiac troponin assays compared to common cut-offs in both men and women undergoing cardiovascular disease risk assessment; 2- the clinical utility of high-sensitivity cardiac troponin assays for cardiovascular disease prevention in women.

The manuscript is concise and clear. Considering both pathophysiological and clinical points of views, the issues discussed in this review are interesting and relevant. I would like to address to Authors some specific points in order to further improve the scientific message of this review.

Specific Points

1.     Introduction, page 1, line 48. Authors should add a full stop in the sentence: “ … system [3-5]. Use of clinically …”.

2.     Introduction, page 2, lines 62 and 63. Two recent expert documents suggest the screening of cardiovascular risk in the general population using the assay of cardiac troponins using the high-sensitivity methods (usually indicated as: hs-cTnI and hs-cTnT) [Farmakis D, et al. Eur Heart J 2020;41:4050-4056; Clerico A, et al. Clin Chem Lab Med 2021; 59: 79-90].

3.     Why is troponin an (almost) ideal biomarker?, page 3, lines 125 and 126. The pathophysiological and clinical relevance of biological variation of hs-cTnI and hs-cTnT methods have been recently discussed in detail [Clerico A et al. Clin Chem Leab Med 2021].

Author Response

GENERAL COMMENT

The aim of this review is to summarize our current understanding of sex-based differences in cardiac troponin levels.  The two most important issues discussed by Authors are: 1- the prognostic significance of sex-specific diagnostic thresholds of high-sensitivity cardiac troponin assays compared to common cut-offs in both men and women undergoing cardiovascular disease risk assessment; 2- the clinical utility of high-sensitivity cardiac troponin assays for cardiovascular disease prevention in women.

The manuscript is concise and clear. Considering both pathophysiological and clinical points of views, the issues discussed in this review are interesting and relevant. I would like to address to Authors some specific points in order to further improve the scientific message of this review.

I found this review interesting and well organized. Figure is very clear and exhaustive.

Author’s Reply: Thank you for taking the time to review our manuscript.

SPECIFIC COMMENT

  1. Introduction, page 1, line 48. Authors should add a full stop in the sentence: “ … system [3-5]. Use of clinically …”.

Author’s reply: item verified and modified accordingly.

  1. Introduction, page 2, lines 62 and 63. Two recent expert documents suggest the screening of cardiovascular risk in the general population using the assay of cardiac troponins using the high-sensitivity methods (usually indicated as: hs-cTnI and hs-cTnT) [Farmakis D, et al. Eur Heart J 2020;41:4050-4056; Clerico A, et al. Clin Chem Lab Med 2021; 59: 79-90].

Author’s reply: item verified and modified accordingly. Suggested studies were thoughtfully included in the manuscript.

  1. Why is troponin an (almost) ideal biomarker?, page 3, lines 125 and 126. The pathophysiological and clinical relevance of biological variation of hs-cTnI and hs-cTnT methods have been recently discussed in detail [Clerico A et al. Clin Chem Lab Med 2021].

Author’s reply: item verified and modified accordingly. The suggested study was thoughtfully included in the manuscript.

Reviewer 3 Report

This literature review is devoted to a very interesting and actual topic.

The use of troponin as a marker of myocardial damage, on the one hand,
contributed to the improvement of the diagnosis of myocardial infarction, but,
on the other hand, caused a large number of questions,
because its increase is associated not only with myocardial infarction.
And this review the authors presents these reasons. And I consider, that the literature analyzed by the authors on the features
of changes in the level of troponin namely in women, is very important.
Indeed, the clinical manifestations of ischemia in women have their own characteristics
and it is important that changes in troponin levels are also associated with sex.
The review presents the results of studies,
demonstrating how important it is to continue
to study the criteria for cardiovascular risk in women,
which may be different from the male population.

Author Response

GENERAL COMMENT

This literature review is devoted to a very interesting and actual topic. The use of troponin as a marker of myocardial damage, on the one hand, contributed to the improvement of the diagnosis of myocardial infarction, but, on the other hand, caused a large number of questions, because its increase is associated not only with myocardial infarction. And this review the authors presents these reasons. And I consider, that the literature analyzed by the authors on the features of changes in the level of troponin namely in women, is very important. Indeed, the clinical manifestations of ischemia in women have their own characteristics and it is important that changes in troponin levels are also associated with sex. The review presents the results of studies, demonstrating how important it is to continue to study the criteria for cardiovascular risk in women, which may be different from the male population.

Author’s Reply: Thank you for your appreciation and for taking the time to review our manuscript.